# The Load Shifting Potential of Domestic Refrigerators in Smart Grids: A Comprehensive Review

Luís Sousa Rodrigues [ID], Daniel Lemos Marques [ID], Jorge Augusto Ferreira, Vítor António Ferreira Costa [ID], Nelson Dias Martins *[ID] and Fernando José Neto Da Silva [ID]

Centre for Mechanical Technology and Automation, Mechanical Engineering Department, University of Aveiro, 3810-193 Aveiro, Portugal
* Correspondence: nmartins@ua.pt

**Abstract:** Domestic refrigeration and freezing appliances can be used for electrical load shifting from peak to off-peak demand periods, thus allowing greater penetration of renewable energy sources (RES) and significantly contributing to the reduction of $CO_2$ emissions. The full realization of this potential can be achieved with the synergistic combination of smart grid (SG) technologies and the application of phase-change materials (PCMs). Being permanently online, these ubiquitous appliances are available for the most advanced strategies of demand-side load management (DSLM), including real-time demand response (DR) and direct load control (DLC). PCMs are a very cost-effective means of significantly augmenting their cold storage capacity and, hence, their load-shifting capabilities. Yet, currently, refrigerators and freezers equipped with PCMs for DSLM are still absent in the market and research works focusing on the synergy of these technologies are still scarce. Intended for a multidisciplinary audience, this broad-scoped review surveys the literature to evaluate the technological feasibility, economic viability and global impact of this combination. The state-of-the-art of SG-enabling technologies is investigated—e.g., smart meters, Internet-of-Things (IoT)—as well as current and future standards and norms. The literature on the use of PCMs for latent heat/cold storage (LHCS) is also reviewed.

**Keywords:** domestic refrigerator-freezer; cold thermal energy storage (CTES); phase-change materials (PCMs); smart grids (SGs); peak-load shifting (PLS); demand-side load management (DSLM); demand response (DR); communications-based demand response (CBDR); advanced metering infrastructure (AMI); greenhouse gases (GHG) emission mitigation



## 1. Introduction

Domestic refrigerators have the potential to be used for demand-side load management (DSLM) and demand response (DR), due to their inherent capacity to store cold and, hence, postpone electric energy consumption to low energy demand periods [1]. Contrary to other household electrical appliances, they are permanently available for this purpose. This may not only result in immediate electricity bill savings, but also has the potential to contribute to decreasing the peak load on the grid [2]. This capability can be more effectively harnessed with: (a) connectivity to an SG framework and (b) the application of phase-change materials (PCMs) for enhanced latent cold storage (LCS) capacity [3–5].

This review focuses specifically on the current technological feasibility and economic viability of integrating household refrigerators and chest freezers in a connected smart grid (SG paradigm, as well as on the potential systemic impact of the widespread adoption of this approach. Evaluating both the economic viability and technological feasibility is, necessarily, a multidisciplinary endeavor. Thus, this review is broadly scoped and intended for a broad readership, comprising macro- and micro-economic impacts (i.e., from systemic- to consumer-level) and several technological and engineering fields. Each subject is covered at an introductory scientific and/or technological level, with just enough depth to achieve

the stated goal. For a more in-depth analysis, the reader is encouraged to explore the numerous references provided. Figure 1 shows a general overview of the covered topics.

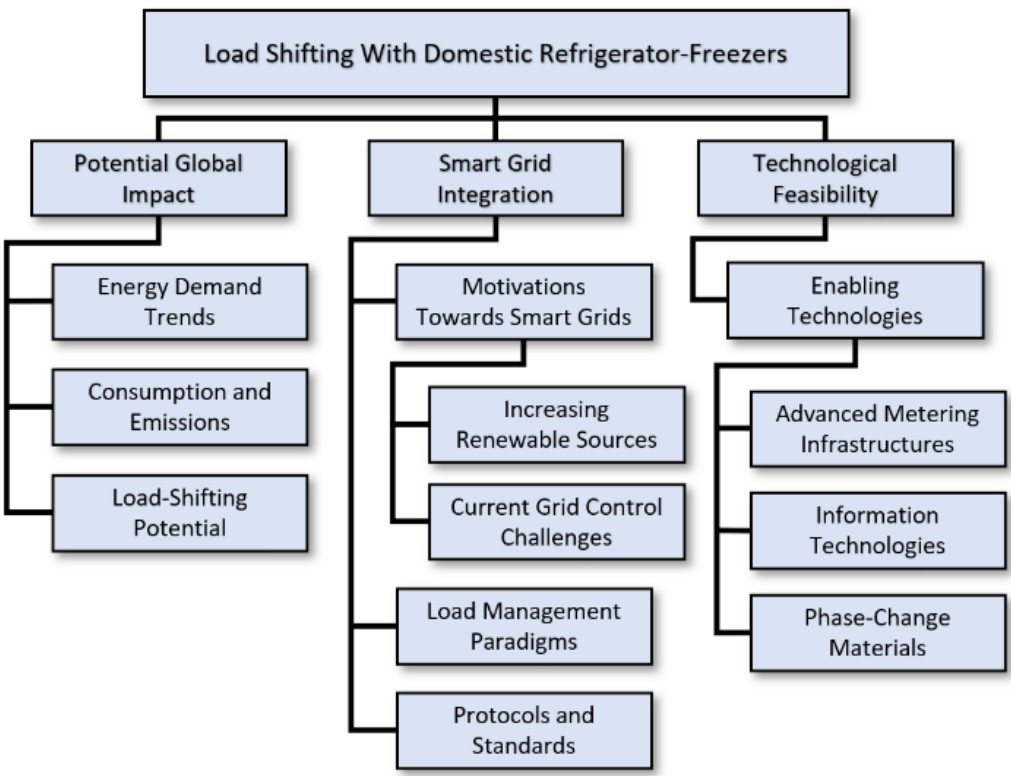

**Figure 1.** Overview of the subjects covered in the present literature review.

Harmonization between energy production and demand is becoming increasingly difficult to achieve, as the contribution of renewable energy sources (RES) is becoming more important [6]. Unfortunately, the profile of renewable generation is rarely in phase with demand, giving rise to supply deficits and excesses (Figure 2). RES production fluctuations can be regularized with enough storage capacity. Unfortunately, viable solutions for the needed storage capacity are still lacking. Hence, the importance of augmenting the ability to control the grid from the demand side [6].

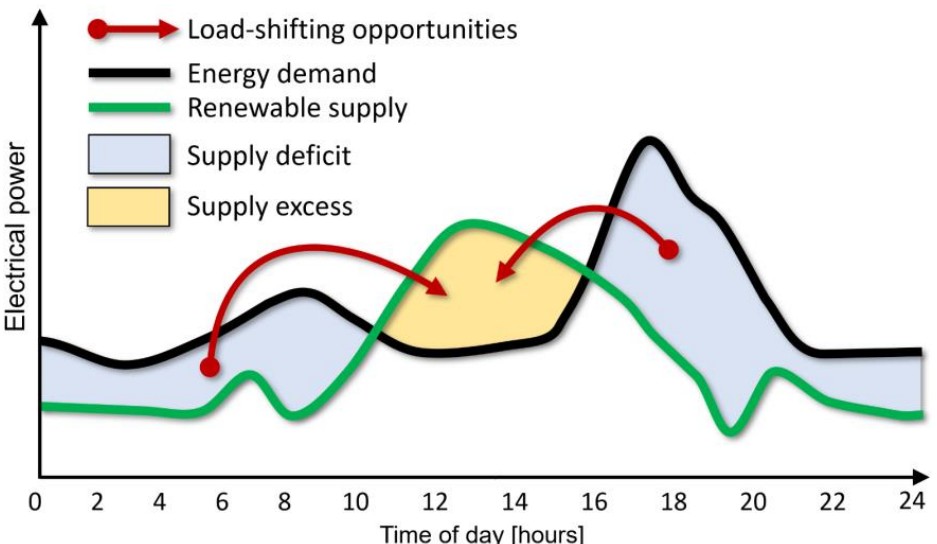

**Figure 2.** Typical daily renewable supply and demand profiles, and load-shifting opportunities.

Instead of relying solely on the supply side, SG approaches allow the active participation of the energy consumers in the task of grid regularization. Thanks to the advances in information and communication technologies (ICT) and their widespread availability, it is now feasible to respond to demand–supply imbalances—from the demand side—in real-time, without the need for human intervention [6].

The extensive adoption of SG technologies—particularly DSLM/DR and distributed small-scale storage—is virtually certain, mainly due to: (a) growing energy demand (especially in developing countries), (b) ageing infrastructures (especially in developed countries), and (c) increasing penetration of RES in the energy mix, stimulated by pressing greenhouse gases (GHG) emissions reduction objectives and the rising prices of fossil fuels [6,7]. If this was true some months ago, its relevance was strongly increased by the energy crisis associated with the military action of Russia on Ukraine.

PCMs can store a considerable amount of thermal energy and are very competitive when compared to electrochemical batteries: they are cheaper, long-lasting, and environmentally friendlier. This explains the large number of studies focusing on PCMs for thermal energy storage: a cursory search easily produces numerous results (on the order of hundreds of thousands). PCMs are particularly adequate for application in domestic refrigerators since they also help to (a) stabilize the cold box temperature (which is crucial for the preservation of goods) and (b) increase energy efficiency (as demonstrated by numerous studies). Furthermore, they increase the resilience to energy supply interruptions—which, unfortunately, are becoming more frequent, not only in countries with unreliable grids but, also, in developed regions (as demonstrated by recent events in the USA, specifically in California and Texas).

Vapor compression cycle refrigeration apparatuses with enhanced LCS capacity are particularly adequate for DSLM/DR as the idle time of the compressor can be considerably prolonged [8,9]. By harnessing the full potential of PCMs, it can be realistically speculated that close to 100% of the energy used for cooling, refrigeration, and freezing in a household, can be shifted to off-peak periods [1,10]. Figure 3 illustrates the basic connective model of the reviewed technologies.

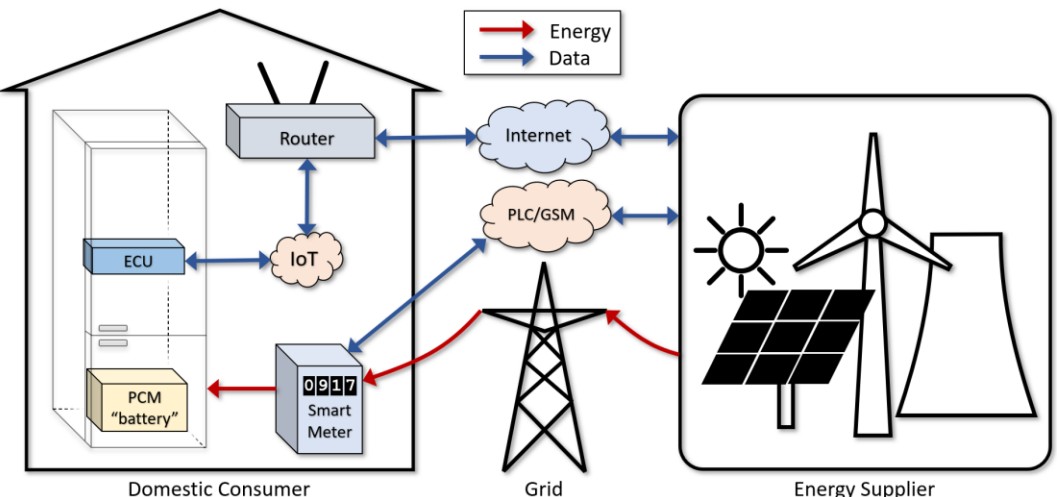

**Figure 3.** Topological overview of the reviewed technologies.

### 1.1. Contents' Structure

To aid in navigating the content, Table 1 outlines the paper's structure. An integral reading is not necessary to grasp the main findings and conclusions. At the end of the two largest and more in-depth sections (*2. Phase-Change Materials* and *3. Smart Grids: Motivations, Goals and Paradigms*) the reader can find short summaries, and, before the *Conclusions*, there is a concise *Discussion* section.

**Table 1.** Contents outline.

*1.2. A Brief Critical Review of Some Relevant Works*

Although there are abundant studies on the application of PCMs to domestic refrigerators and/or freezers, the vast majority focus almost exclusively on enhancing efficiency (with mixed results and, sometimes, not very significant gains in energy efficiency and/or energy consumption reduction). A subset also looks at temperature stability, and/or autonomy. There are very few works specifically aimed at the interesting potential of using PCMs for purposes of DSLM/DR or peak-load shifting/shaving. Studies that look at the promising synergy with SGs are even rarer.

In 2007, Stadler et al. compared two load-shifting control strategies with "adaptive fridges" by simulating 5000 controllable refrigerators that respond to external commands or signals [11]. They concluded that thermal storage with large numbers of refrigerators is "well suited for short-time balancing". To extend the response periods, the authors suggest the inclusion of other thermal apparatuses (e.g., electric boilers, off-peak storage heaters, and freezers). However, their models did not contemplate or mention using PCMs to extend cold storage capacities.

In 2012, Zehir and Bagriyanik modelled the potential effects of performing DSM with domestic refrigeration and freezing equipment [12]. Basing their calculations on real measurements, they estimated that 37.9% of the load could be shifted from peak to off-peak periods (using an inefficient class-G refrigerator, representative of the most common type— more than 60%—in use in the USA). It is legitimate to speculate that the percentage of load shifting could be much higher if the studied model was equipped with PCMs to increment the cold storage capacity. In a multiple rate tariff scheme with an off-peak discount of 5.4%, the yearly savings amounted to 11.4%.

Another 2012 study by Kornbluth et al. investigated the economic viability of load shifting in California and Denmark, using a freezer equipped with a passive PCM installation (i.e., the PCM is simply placed in contact with the evaporator, without any type of active thermal transfer control) [13]. Kornbluth et al. calculated a production cost of about USD 50.00 per appliance. No SG integration was investigated: only load shifting from peak to off-peak, with time-of-use (ToU) multi-rate pricing. Accounting for energy prices at the time, the authors concluded that a Californian consumer would reach the break-even point

within six years. In Denmark, however, the solution did not make economic sense at the time (results may considerably differ presently and/or in the future).

Still, in 2012, Baghina et al. presented a predictive control strategy for implementing real-time DR with a domestic freezer [14]. This type of work is relevant for simulating and/or modelling the thermal behavior of cooling/freezing appliances. However, predictive algorithms have important limitations for real-time control since the machine needs to react to continuously changing conditions and predominantly stochastic events (e.g., door/lid openings, loading, and unloading of hot/cold goods, etc.). It also must always prioritize the preservation of goods, i.e., it must turn on cold generation (or retrieval from the PCM) when the cold box temperature reaches an allowable maximum limit, regardless of energy price or the balance between supply and demand at the time.

In 2013, Lakshmanan et al. also presented a "simple temperature prediction strategy for the evaluation of the expected power consumption" for domestic refrigerators [15]. This work is also relevant as a basis for simulating/modelling refrigerators/freezers. However, these models have significant limitations for control purposes, as aforementioned.

Additionally, in 2013, Taneja et al. presented an interesting and significant "flexible and responsive" domestic refrigerator prototype equipped with actively controlled LHCS and wireless communication [1]. This is one of few works specifically focused on household refrigerators equipped with both actively controlled cold storage and an Internet connection, fully exploitable for DR in an SG context. The authors investigated two load DR management modalities: price-responsive (or real-time pricing, RTP) and supply-following. With their particular setup, they managed a total autonomy of over 6 h (although with a detrimental 13% energy consumption increase). In a more practical/typical situation, however, the authors claim a reduction of up to 13% in running costs in the summer. Furthermore, it is claimed that "at higher levels of renewables penetration, with 20% of refrigerators in California adopting this technology ( . . . ), flexible refrigerators can shave off 5% of peak capacity needs."

Again in 2013, Zamboni et al. presented a controller prototype to enable SG/DSLM/DR functionalities in a commercial refrigerator model (without any additional LHCS capabilities) [16]. Their control algorithm also has some predictive capabilities, allowing some adaptation to the consumer's habits. Zamboni et al. claim average energy cost savings of 51% by shifting consumption from demand peak to off-peak periods.

In 2014, "cold storage devices for smart grid integration" were studied by Waschull et al. [17]. The study aimed at maintaining positive temperatures in the range of 4 °C to 10 °C. The authors found that using a eutectic $KHCO_3/H_2O$ solution with a phase change temperature of about $-6$ °C was most effective for this purpose. The most innovative aspect of this work was the investigation of possible state-of-charge (SoC) sensors for the PCM. Electrical, optical, and volumetric devices were evaluated. The authors found that the best type of sensor was a conductive silicone membrane. Another innovative aspect of this work was the use of an actively controlled thermosyphon (in contrast with most other works that only look at passive solutions). Wi-Fi connectivity was used for control and telemetry, going a step further in potential SG integration than most other studies. These authors also mention the interesting possibility of using this technology for direct solar-powered refrigeration.

Still in 2014, Almenta et al. presented a smart load response model for a refrigerator-freezer [18]. This model aims to reduce the energy cost to the owner by regulating the refrigerator-freezer thermostat according to three distinct strategies: (a) peak shaving and valley filling (i.e., shifting consumption from peak to off-peak periods); (b) "real-time" energy price signal; and (c) wind availability. It is claimed that the model's implementation could provide 54 MWh of valley filling (consumption in low demand periods) and reduce energy costs by 55% during peak periods in Ireland.

Additionally, in 2014, Lakshmanan et al. presented a method for estimating the amount of energy that can be shifted in a domestic refrigerator for DSLM/DR purposes [19]. This straightforward method only requires knowledge of two variables: cool box temperature and compressor energy consumption.

In 2015, Sonnenrein et al. conducted a study with a similar passive PCM setup, using copolymer-bonded PCMs [8]. Besides their results showing smaller consumption, better stability, and uniformity in the spatial distribution of temperatures, they also briefly evaluated the potential for DSLM (albeit without SG integration). Their specific setup was able to postpone consumption for up to 1.8 h. No economic impact analysis was performed.

Also published in 2015, another study was conducted by Barzin et al. [9]. The impact of price-based peak load shifting (outside the SG context) with a freezer equipped with PCMs was investigated. Cost savings of up to 16.5% were recorded for a domestic freezer.

Still, in 2015, Almenta et al. presented an "aggregated" peak shaving and valley filling control strategy for refrigerator-freezers, claiming benefits for Ireland's electricity market: a potential peak demand reduction of 40 MW to 55 MW and an increase in off-peak demand of up to 30 MW [20].

In 2016, Mathias et al. defended a distributed grid control model with the generalized implementation of smart loads (including refrigerators) [21]. The authors used a demand dispatch paradigm—i.e., a one-way communication between the energy supplier and the automated appliances on the consumer side, with the load management decisions being preponderantly made on the demand side.

A comprehensive review of DSM in refrigeration applications by Arteconi and Polonara (2017) is worth mentioning here [22]. Besides refrigerators, it covers heat pumps (for space and water heating) and air conditioning. The authors surveyed the available literature and found significant potential for realizing DSLM with refrigeration appliances, particularly using PCMs.

A recent study by Maiorino et al. (2020) demonstrated the viability of achieving running cost savings with a prototype refrigerator equipped with a passive-type PCM storage by shifting the load to off-peak periods [23]. This was accomplished with ToU-type electricity tariffs across different European countries, either with two or three differentiated tariff slots (i.e., 2-ToU and 3-ToU). The authors proposed an algorithm for the optimization of energy bill savings. In some circumstances, they found that the inclusion of PCMs can actually increase total energy consumption (even leading to slightly increased running costs in the most unfavorable cases). However, the increased autonomy awarded by the PCM storage capacity allowed a greater shifting of consumption to lower tariff periods, generally with a net reduction in running costs. In an optimized scenario, they achieved a reduction of up to 35% in energy costs in France. Typical savings for Spain were around 21%. The control algorithm accounting for the ToU achieved cost savings in all cases, without using PCMs. However, the possibility of SG integration and its potential systemic impact was not considered, nor were the benefits of greater control offered by active PCM approaches.

Table 2 shows a summary of the findings of the most relevant studies found and their main limitations.

**Table 2.** Most relevant results from studies on load-shifting with domestic refrigerators-freezers.

| Ref. | Author(s) | Year | Studied Subject | Main Findings/Claims | Comments/Limitations |
|------|-----------|------|-----------------|----------------------|----------------------|
| [11] | Stadler et al. | 2007 | Load-shifting simulation of 5000 refrigerators responding to external commands. | Thermal storage with large numbers of refrigerators is well-suited for short-time balancing. | Did not consider the application of PCMs. Unsatisfactory real-world quantification of results. |
| [12] | Zehir and Bagriyanik | 2012 | Modeling consumer effects of DSM with refrigerator-freezers. | 37.9% load-shifting was achieved, without the use of PCMs. Annual cost savings of 11.4% with an off-peak discount of 5.4%. | Did not include the use of PCMs. Only ToU tariffs were considered. No further SG integration was considered. |

**Table 2.** *Cont.*

| Ref. | Author(s) | Year | Studied Subject | Main Findings/Claims | Comments/Limitations |
|---|---|---|---|---|---|
| [13] | Kornbluth et al. | 2012 | The economic viability of load shifting with a freezer equipped with PCM (in California and Denmark). | Cost of USD 50.00 calculated per appliance. Break-even point in 6 years in California. Not economically viable in Denmark (in 2012). | Only considered ToU tariffs. No further SG integration was considered. |
| [14] | Baghina et al. | 2012 | Predictive control strategy for real-time DR with a domestic freezer using PCMs. | Demand response is viable with predictive algorithms. | Lack of reliable quantification of results. Limitations of the predictive algorithm concerning its real-world application. |
| [1] | Taneja et al. | 2013 | Domestic refrigerator prototype equipped with actively controlled LHCS with PCM and wireless communication. | Autonomy over 6 h. Reduction of up to 13% in running costs. Capacity to shave 5% of peak load with market adoption of 20% in California. | A very comprehensive study. Two demand-response models were investigated: price-responsive and real-time supply-following. Very significant results. |
| [16] | Zamboni et al. | 2013 | Controller prototype enabling SG/DSLM/DR in a commercial refrigerator model (without PCMs). | Cost savings for the consumer of up to 51% claimed. | Predictive and adaptive control algorithm (including adaptation to consumer habits). PCMs not considered. |
| [17] | Waschull et al. | 2014 | Cold storage devices for SG integration. | $KHCO_3/H_2O$ eutectic solution was found to be the ideal PCM for maintaining temperatures in the 4° to 10 °C range. Silicone membrane was found to be a good state-of-charge (SoC) device for a PCM. | Actively controlled "thermosyphon" and wireless connectivity for control and telemetry. Relevant study of several mechanisms for measuring the state-of-charge (SoC) of the PCM. Only positive temperatures (4 °C to 10 °C) were considered. |
| [18] | Almenta et al. | 2014 | Smart load response model for a refrigerator-freezer, allowing peak-shaving and valley-filling, responding to real-time pricing and wind availability. | 54 MWh of valley-filling claimed with wide adoption in Ireland. 55% running-costs reduction in Ireland claimed. | Quantification of results confined to Ireland. The control mechanism is limited to the regulation of the thermostat's set-point. |
| [8] | Sonnenrein et al. | 2015 | Passive application of polymer-bonded PCMs to domestic refrigerator-freezers. | Load-shifting up to 1.8 h. | Potential for DSLM (briefly) evaluated. No SG integration was considered. No quantification of economic impact was done. |
| [9] | Barzin et al. | 2015 | Impact of price-based peak load shifting for a domestic freezer with PCMs. | Running cost savings of up to 16.5% were recorded. | Only ToU pricing was evaluated. No further SG integration was considered. |

**Table 2.** *Cont.*

| Ref. | Author(s) | Year | Studied Subject | Main Findings/Claims | Comments/Limitations |
|---|---|---|---|---|---|
| [20] | Almenta et al. | 2015 | Peak shaving and valley filling control strategy for refrigerator-freezers. | Potential peak demand reduction between 40 MW and 55 MW. Increase in off-peak demand of up to 30 MW. | Quantification of results confined to Ireland. Only systemic effects were studied. No quantification of savings in running costs for the consumer. Use of PCMs not considered. |
| [23] | Maiorino et al. | 2020 | Prototype refrigerator with passive-type PCM storage and control algorithm for running cost optimization. | Up to 35% reduction in running costs in France and 21% in Spain. | A comprehensive study including 2 and 3 ToU tariff slots, across two European countries. No further SG integration was considered. |

*1.3. Global Context*

1.3.1. Energy Demand Trends

Until 2019, energy demand had been steadily increasing, and this trend was consensually forecasted to continue (and even accelerate), mainly due to the emerging economies of developing countries. This demand increase stresses the existing infrastructures, incentivizing the investment in SG technologies [24,25].

The recent COVID-19 pandemic has momentarily broken this tendency. In 2012, the International Energy Agency (IEA) forecasted a 12% worldwide growth in energy demand for 2020 in comparison to 2010 [10,17]. In 2020, the IEA reassessed its forecasts and estimated a drop in global energy demand of about 5% (for the year). However, global electricity demand would only be down by a "relatively modest" 2% [26].

According to general expectations, the global electricity demand rebounded in 2021, growing by 6%, according to the IEA's Electricity Market Report. However, in its most recent July 2022 update, the IEA projects a growth of only 2.4% in 2022, a figure more in line with the five years preceding the COVID-19 pandemic [27]. This figure reflects the global slowdown in economic growth, mainly caused by the soar in energy prices following the invasion of Ukraine by Russia.

Despite the reported uncertainty of the most recent projections, the drop in energy demand triggered by the war in Ukraine is transient and unlikely to deter investment and research in renewables and energy utilization efficiency. Global energy demand is still projected to continue growing for the foreseeable future and the urgency of reducing GHG emissions has not diminished. Moreover, since the residential sector is one of the main drivers of this growth—greatly thanks to developing countries and increasing urbanization—achieving the global goal of reducing emissions will, necessarily, include the participation of the small residential consumers [5,28–30].

1.3.2. Potential for Thermal Energy Storage (TES) in The Residential Sector

Currently, the residential sector is responsible for about 30% to 40% of the total worldwide energy consumption [31]. Electrical loads with thermostatic control—e.g., refrigeration, freezing, cooking, water heating, washing machines, laundry drier, space heating and cooling—constitute the largest percentage of residential energy consumption, exceeding 50% [32,33]. If TES technologies (e.g., PCMs) are applied to these loads, 15% to 50% of the energy consumption—depending on the type of load—could be, realistically, shifted from peak to off-peak with potential systemic efficiency improvements [34,35].

1.3.3. Domestic Refrigeration: Global Impact on Energy Consumption and Emissions

Even though the impact of a refrigerator does not seem, a priori, particularly important in the total energy bill of a typical household, its cumulative weight at national, transnational, and continental levels, should not be underestimated. In particular, if its prevalence

is accounted for (refrigerators are present in more than 89% of EU households [8,36–38]). On average, 13.4% of (all) residential energy consumption in member states of the OECD comes from the cooling and freezing of foodstuff [8]. In Germany, that percentage is about 20%, corresponding to about 7% of the national total energy budget [8].

The International Institute of Refrigeration (IIR) estimates that the refrigeration sector consumes about 17% of the electricity consumed worldwide, of which 45% comes from the residential sector, i.e., almost 8% of the total electricity produced [22,39]. In European countries, these percentages are slightly lower (presumably due to colder climates and more efficient appliances). For example, refrigeration (domestic, commercial, and industrial) accounted for 14% of the electric energy consumption in Germany in 2009. Of the total annual 71 TWh, 24 TWh were used by domestic refrigerators, which is about 5% of the total electric energy produced [17].

It is estimated that, in 2016, there were 1.4 billion domestic refrigerators and freezers worldwide, each with an average energy consumption of 450 kWh/year. These appliances are responsible for about 14% of the energy consumed by the residential sector, causing a yearly emission of about 450 million $CO_2$ tons [40,41].

The annual world production of domestic refrigerators was about 80 million units in 2009. This is about twice the numbers of the mid-1990s [42], corresponding to an average yearly growth of about 6.5%. Reliable production statistics for the last decade are difficult to gather, but it is reasonable to assume that these growth rates have been, at least, maintained. Barring the transient effects of the COVID-19 pandemic and the conflict in Ukraine, the global yearly GDP growth since 2010 has been reasonably stable at about 3%, and India's and China's at least double that, according to The World Bank [43]. This means that, for 2021, the yearly production of refrigerators should be, at least, about 160 million.

The global number of refrigerators in use was estimated to be, approximately, 1 billion in 2008 and 1.4 billion in 2016 [40,42], which corresponds to a yearly growth rate of, approximately, 4.5% (the 2% discrepancy can be explained by some inevitable obsolescence). Considering this growth rate, this number may be now between 1.7 billion and 2 billion (potentially closer to the latter). Assuming the most conservative figure of 1.7 billion (reflecting the recent drop in the world's GDP) and the improvement in the efficiency of more recent equipment, it is estimated that refrigerators and freezers should currently account for the annual emission of over 540 million $CO_2$ tons. According to the most recent numbers from the IEA, this represents about 1.7% of the world's total emissions in 2020 [44].

1.3.4. The Potential Impact of Domestic Refrigerators on Peak Load Shifting

In Germany, refrigerators and freezers account for a combined total power of 3.6 GW [10]. Using latent cold storage with PCMs and efficient control algorithms, all this power would be (theoretically) available for storage—allowing, for instance, the installation of an additional 3.6 GW solar wind farm, without any significant negative impact on grid stability [10]. Even without resourcing to cold storage with PCMs, it is estimated that the potential impact of all domestic refrigerators in Germany for load management can be up to 800 MW [33].

As mentioned before, Zehir and Bagriyanik (2012) calculated that, even without using latent cold storage, about 40% of electrical load can be postponed to off-peak hours [12]. If the refrigerator is equipped with LHCS, the peak load shifting (PLS) percentage should be higher. Nevertheless, keeping estimates conservative, 40% will be considered a reasonably attainable base percentage. In round figures, if 50% of worldwide refrigerators could shift 40% of their consumption to RES, annual emissions of more than 100 million $CO_2$ tons could be averted.

## 2. Phase-Change Materials

### 2.1. Storing Cold with PCMs: Underlying Physics

Figure 4 shows the typical temperature evolution of a liquid–solid PCM going through a complete freezing–thawing cycle, where $T_f$ is the melting temperature. As can be observed,

the temperature remains essentially constant at $T_f$ during the liquid–solid and solid–liquid phase transitions (where both liquid and solid phases co-exist in varying proportions).

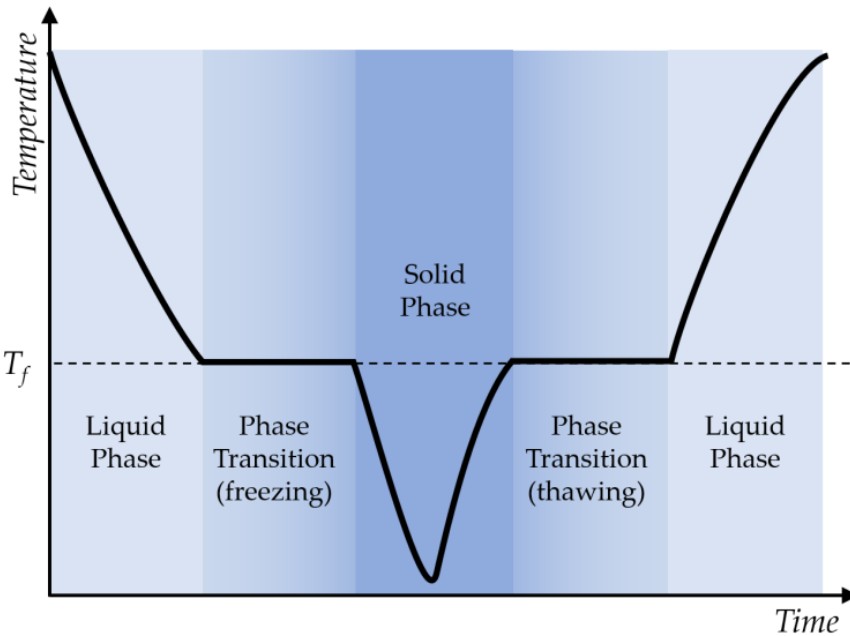

**Figure 4.** Temperature evolution of a liquid–solid PCM during a freezing–thawing cycle.

The heat released or absorbed by the PCM does not translate into temperature changes during the phase transitions, contrary to when the PCM is fully liquid or solid. Hence, the expression "latent heat" (versus "sensible heat"). This occurs because any heat released or absorbed by the PCM is "used" to form or break intermolecular bonds, respectively, rather than translating into molecular kinetic energy (i.e., temperature). The latent heat exchanged during phase transitions is given by

$$Q|_{x_1}^{x_2} = \int_{x_1}^{x_2} mh_f \, dx, \text{ with } 0 \le x_i \le 1, \tag{1}$$

where $m$ is the mass of the PCM, $h_f$ is its specific enthalpy of fusion and $x$ is the liquid fraction. Since both $m$ and $h_f$ are constants, the expression simply becomes

$$Q|_{x_1}^{x_2} = mh_f(x_2 - x_1), \text{ with } 0 \le x_i \le 1. \tag{2}$$

Thus, considering a complete phase-change transition ($x$ changing from 0 to 1 or vice versa), a mass $m$ of a given PCM with a specific enthalpy of fusion $h_f$ can store or release an amount of heat given by:

$$|Q| = mh_f. \tag{3}$$

On the other hand, the sensible heat exchanged outside phase transitions (i.e., when only one phase is present) between two temperatures $T_1$ and $T_2$ is given by

$$Q|_{T_1}^{T_2} = \int_{T_1}^{T_2} m \cdot c_P(T) \, dT, \tag{4}$$

where $c_P$ is the constant-pressure specific heat of the PCM, which, in general, depends on the temperature $T$. For the majority of practical cases, however, $c_P$ can be assumed constant (the changes are usually too small to be significant within most practical temperature ranges). Equation (4) then simplifies to

$$Q|_{T_1}^{T_2} = mc_P(T_2 - T_1). \tag{5}$$

Considering a case where $T_1$ and $T_2$ are, respectively, below and above the melting point $T_f$ (as shown in Figure 5), the total heat absorbed by the PCM is thus given by

$$Q|_{T_1}^{T_2} = m\left[c_{Ps}\left(T_f - T_1\right) + h_f + c_{Pl}\left(T_2 - T_f\right)\right], \qquad (6)$$

where $c_{Ps}$ and $c_{Pl}$ are the specific heats of the solid and liquid phases, respectively. Equation (6) is equally applicable to the reverse case—i.e., the transition from liquid to solid—the only difference being the sign of $Q$ (i.e., it will be negative in the latter case).

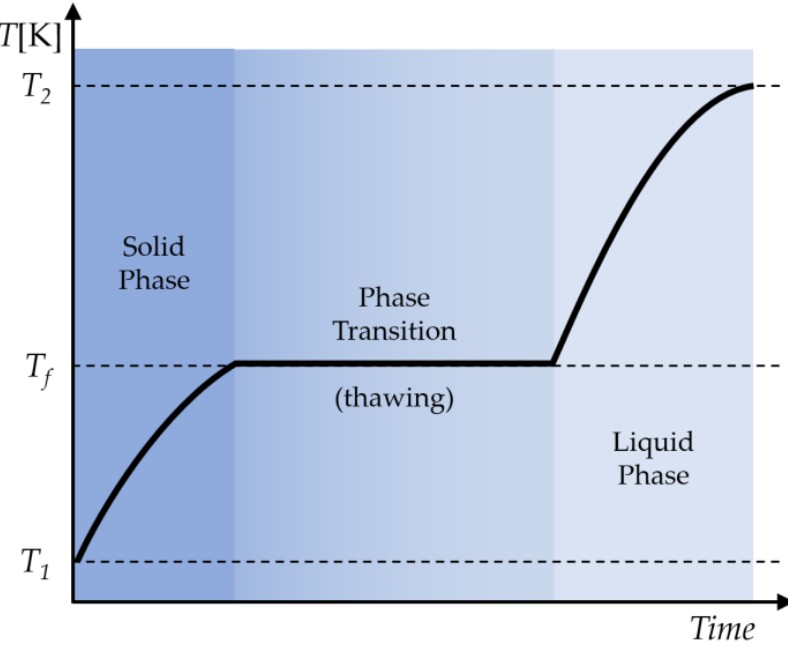

**Figure 5.** Temperature evolution of a PCM, from solid to liquid, including sensible heating outside the phase transition region.

Since $h_f \gg c_P$ (typically, about two orders of magnitude higher) the latent heat stored or released during the phase change is—in most practical applications of PCMs—much greater than the sensible heat exchanged in the purely solid and/or liquid phases (becoming even more important as the difference $T_2 - T_1$ decreases). Hence, the suitability of PCMs for TES.

### 2.2. Phase-Change Materials for Latent Cold Storage

There is a very substantial body of research focusing on the application of PCMs to domestic refrigerators/freezers. The majority aims at improving energy efficiency and some at strengthening the resilience to power supply intermittencies. However, PCMs can also significantly increase the load shifting and demand response capabilities of these appliances. Their main virtues in this application are:

1.  High thermal energy storage capacity per unit mass (comparable and/or competitive with electrochemical batteries), thanks to the considerable difference in entropy (and, hence, enthalpy) between the two (solid and liquid) phases;
2.  Temperature stabilization enhancement, since the temperature of the PCM during the phase transition remains practically constant (see Figure 4).

Oró et al. (2012), Selvnes et al. (2020) and Ghodrati et al. (2022) conducted comprehensive surveys and comparative studies of some of the relevant thermophysical properties of the main classes of PCMs used for LCS [45–47]. Figure 6 illustrates the temperature usability ranges of some of the PCM types identified by these and some other authors [45–52].

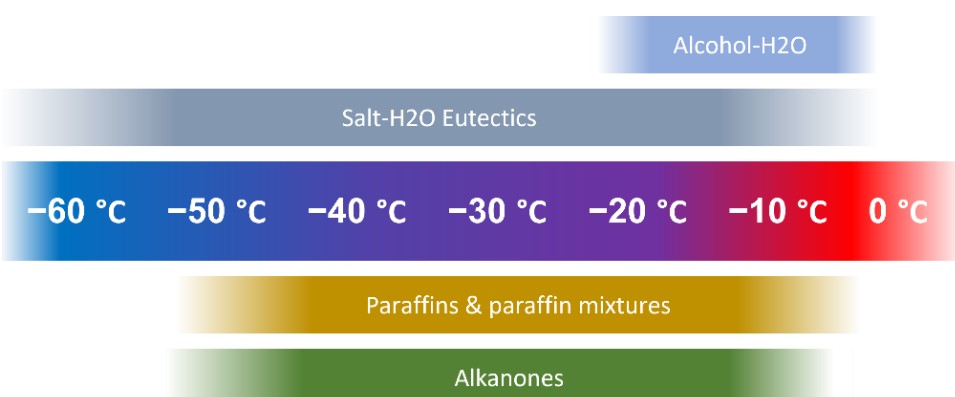

**Figure 6.** Usable temperature ranges for some of the most relevant PCM types for negative Celsius temperatures [45,47–49,51,53].

Of the PCM types shown in Figure 6, the eutectic aqueous salt solution is the most used [3], accounting for the sheer number of publications and the fact that it is commercially and readily available for application in freezers (domestic or not). Likely, the preponderant reason for this predominance is cost. Other desirable advantages include its low- to non-toxicity, environmental friendliness, non-flammability, and—being inorganic—long-term stability. Since these are water-based, the latent heat of fusion is relatively high, and the thermal conductivity is also good [52].

Using eutectic versus non-eutectic solutions is advantageous from the point of view of long-term dimensional and mechanical stability. The salt concentration is precisely controlled, such that it remains homogenous during the phase transition, i.e., there is no segregation of phases of different concentrations and the overall volume remains more constant than that of pure ice. Eutectic solutions are frequently used in gel form, for ease of application, dimensional stability, leak prevention, and evaporation and/or adsorption mitigation. Unfortunately, in many instances, the gelling agent tends to separate from the solution after several freezing–thawing cycles, degrading these advantages.

One of the disadvantages of aqueous eutectic salt solutions is their hysteretic phase-transition behavior: being water-based, they have a strong tendency to supercool (i.e., cool below the transition temperature without solidifying) [49,54]. This can be detrimental to efficiency in current applications [55].

The other significant disadvantage of salt-water eutectics is their poor compatibility with metals. They promote metal corrosion, either through galvanic processes (i.e., in contact with dissimilar metals) or via oxidation (in the presence of dissolved oxygen) [52]. This affects copper, steel, and aluminum, forcing the adoption of poor heat conductors like stainless steel and/or plastics [1].

### 2.3. Advantages and Desirable Characteristics of PCMs

Kapilan et al. (2021) compiled the recent advances in the application of PCMs for cold storage and comprehensively assessed the main advantages and disadvantages of this technology for different types of PCMs [56].

Using electrochemical batteries as a reference, the following advantages can be mentioned:

- Environmentally friendlier than batteries;
- Much longer lifespan compared to batteries;
- Practically no performance degradation with time;
- Practically unlimited number of charge–discharge cycles;
- Low lifecycle costs;
- Low cost per storage capacity (€/kWh) than any type of battery;
- Good specific storage capacity (J/kg), better than lead-acid batteries;
- Good energy storage density (J/m3), comparable to lead-acid batteries;

- Generally non-toxic;
- Generally easy to dispose of;
- Easy to charge–discharge (no control hardware required);
- Can be fully discharged and charged without degradation.

Figure 7 illustrates how a eutectic PCM (KCl/$H_2O$, in this case) compares with three common types of electrochemical batteries—lead-acid, Nickel-Metal Hydride (NiMH), and Lithium-ion (Li-ion)—in terms of specific storage capacity (Wh/kg) [57,58]. Battery specifications can vary widely, depending on the model, type of use and manufacturer. For instance, Li-Iion batteries can be optimized for specific capacity, discharge current, longevity, safety, etc., which, frequently being conflicting characteristics, result in a wide range of specific capacities, denoted by the light-shaded portion of the bar. Additionally, lead-acid battery specifications depend on their intended application: batteries intended for renewable energy storage are optimized for durability and resilience to full discharge (deep-cycling); car batteries, on the other hand, are optimized for peak maximum current (needed for starting the engine) and are not tolerant to full discharge. Hence, the bar chart is intended only to provide an approximate comparison, merely adequate for the current purposes. The battery data is merely indicative and was obtained from several manufacturers' datasheets and Ibrahim et al. [58]. For simplicity, only the latent heat of fusion of the PCM was accounted for (i.e., sensible heat was not considered). Additionally, no consideration was given to the unavoidable inefficiencies inherent to charge–discharge processes (of either type of energy storage).

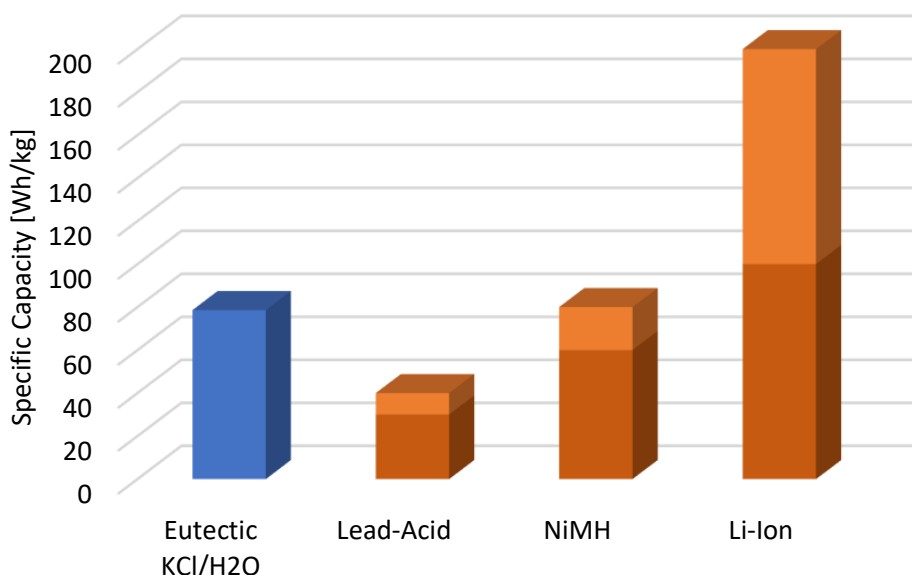

**Figure 7.** Comparison of the specific capacity of a eutectic KCl+H2O PCM with the three more common types of electrolytic batteries [57,59].

As can be seen, in terms of specific capacity (Wh/kg), the aqueous eutectic has, approximately, double the capacity of the very common and inexpensive lead-acid battery. It is roughly equivalent to a NiMH battery and has half the capacity of the best-performing and expensive Li-ion battery. However, when compared with less performant instances of batteries (represented as darker shades of brown in the graph), the advantage of the Li-ion battery is not so clear.

Furthermore, considering that to increase the lifespan of lead-acid and Li-ion batteries, they are not used to their full/nominal capacity, the comparison will be considerably more favorable for the PCM. Additionally, if sensible heat is considered, the merit of the PCM will become even more evident [57]. To illustrate this, Figure 8 shows a more targeted and realistic comparison between the aqueous eutectic KCl/$H_2O$ PCM and the most economical

and common type of electrochemical battery, the lead-acid—in terms of specific capacity (Wh/kg)—when the battery is discharged at only 50% of its nominal capacity to prolong its useful life (which is a more realistic scenario) [60]. Again, charge–discharge efficiencies and sensible heat were not considered. As can be observed, in real-world applications, eutectic aqueous PCMs can store more than three times the amount of energy per unit mass than lead-acid batteries.

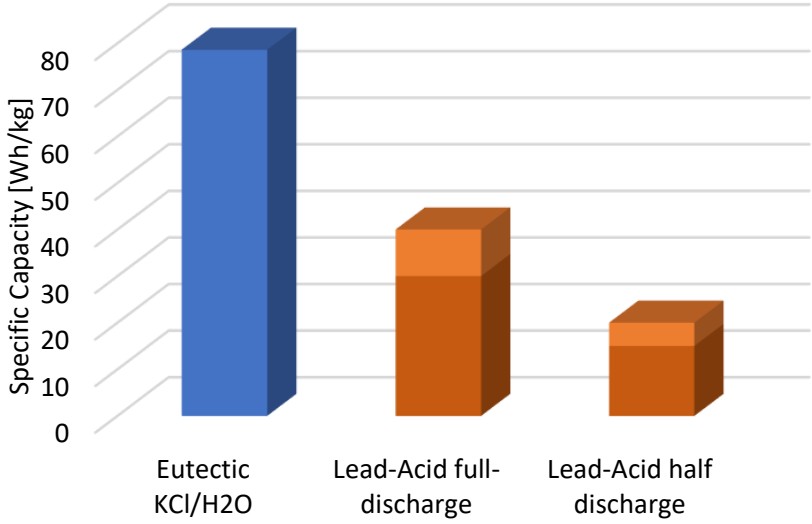

**Figure 8.** Comparison of the specific capacity of a eutectic KCl+H2O PCM with a lead-acid battery [57,59,60].

*2.4. Disadvantages and Practical Problems of PCMs Applied to Refrigerator-Freezers*

Inevitably, PCMs also have disadvantages and problematic and/or undesirable behaviors and characteristics, when considering their application to domestic refrigerator-freezers, such as [56]:

- Poor thermal conductivity;
- Corrosiveness (e.g., aqueous salt eutectics);
- Flammability (e.g., non-aqueous organics: paraffins, alkanones, alcohols);
- Volume variation on phase-change (e.g., non-eutectic aqueous salt solutions);
- Supercooling or phase-change hysteresis (e.g., aqueous PCMs);
- Risk of leakage (e.g., non-gelatinized liquids or unstable gels);
- Phase separation.

Another issue is that—contrary to, e.g., electrochemical batteries—PCM implementations are not "off-the-shelf". They need mechanical construction and integration in situ at the refrigerator factory. Additionally, they require specific design and production engineering [57]. They also require mechanical work from the compressor to be charged. This can be a significant disadvantage if the refrigerator is powered by photovoltaic cells, as there is a minimum power necessary to start the compressor. All the power below this minimum goes to waste unless the solution includes a battery, which can still be charged with little available power. Another disadvantage of PCMs versus electrochemical batteries is the difficulty to measure the current state-of-charge (SoC). However, this does not mean it is not possible. Waschull et al. (2014) investigated some approaches with some success [17]. However, PCMs have, generally, a far smaller specific cost than electrochemical batteries, much longer lifespans, lower maintenance and lifecycle costs, and can be safer, less toxic and more eco-friendly.

Table 3 summarizes the advantages and disadvantages of eutectic aqueous PCMs versus lead-acid batteries [57].

**Table 3.** Advantages and disadvantages of eutectic aqueous PCMs versus lead-acid batteries, for the current application [57].

| | | Eutectic Aqueous PCMs | | Lead-Acid Batteries |
|---|---|---|---|---|
| **Advantages** | ✓ | Cheaper | ✓ | Off-the-shelf |
| | ✓ | Safer/Non-toxic | ✓ | Easier to integrate |
| | ✓ | Eco-friendly | ✓ | Stores electricity |
| | ✓ | Longer lifespan | ✓ | Higher volumetric capacity |
| | ✓ | Higher specific capacity | ✓ | Can charge from low-power sources |
| | ✓ | Low-maintenance | ✓ | Easy to measure the SoC |
| | ✓ | Lower lifecycle costs | | |
| **Disadvantages** | × | Not off-the-shelf (need in-house manufacture) | | |
| | × | Harder to integrate | × | More expensive |
| | × | Require mechanical work (from the compressor) | × | Shorter lifespan |
| | | | × | Toxic |
| | × | Do not store electricity (and cannot convert back into it) | × | More dangerous |
| | | | × | Lower specific capacity |
| | × | Lower volumetric capacity | × | Higher lifecycle cost |
| | × | Difficult to measure the SoC | | |

## 2.5. Practical Implementations of PCMs in Refrigeration Appliances

Some refrigerators equipped with PCMs were already tried on the market [61,62], and some are currently being investigated [57]. Figure 9 shows a practical example of the application of a PCM during the manufacturing process of a prototype chest freezer. The gelatinized PCM—an aqueous eutectic, in this case (colored pale yellow)—is contained in transparent polypropylene sleeves. The sleeves are adhesively affixed to the evaporator tubes.

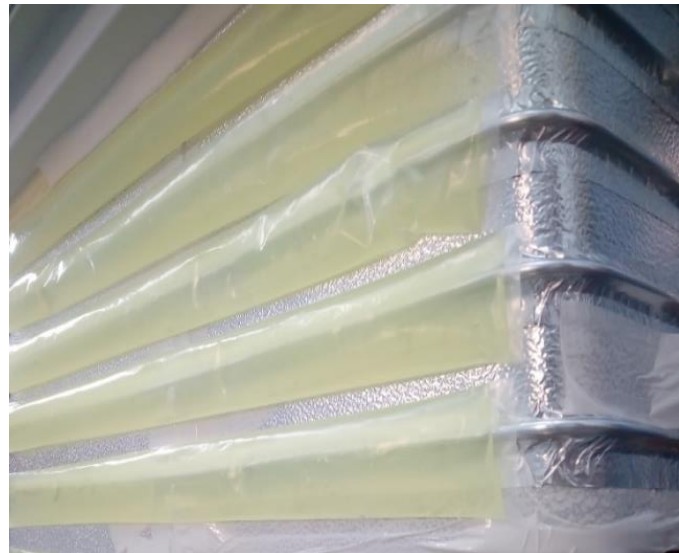

**Figure 9.** Application of a gelatinized eutectic PCM to the evaporator of a prototype chest freezer (photo courtesy: Vítor Silva, Tensai Indústria S.A., Portugal).

The goal is to achieve better efficiency and autonomy by augmenting the thermal inertia of the equipment. The number of compressor on–off cycles decreases, improving efficiency (a full explanation is outside the present scope). The enhanced autonomy not only boosts the load-shifting potential, but also the resiliency to power outages.

In this type of solution, there is no direct control of the "charge-discharge" of the PCM. This can have some detrimental consequences and, in some situations, even slightly decrease the efficiency (a full discussion of the reasons is outside the current scope).

By contrast, in an active implementation—such as the one shown in Figure 10 (similar to one configuration used by Taneja et al. [1])—the "charge-discharge" process can be actively controlled. In this case, the thermal energy transfer between the evaporator and the PCM is moderated by an intermediate heat-transfer fluid, circulated by an electric pump. A microcontroller controls the activation of the circulation pump, so that heat is transferred to and from the PCM only when it is advantageous to do so.

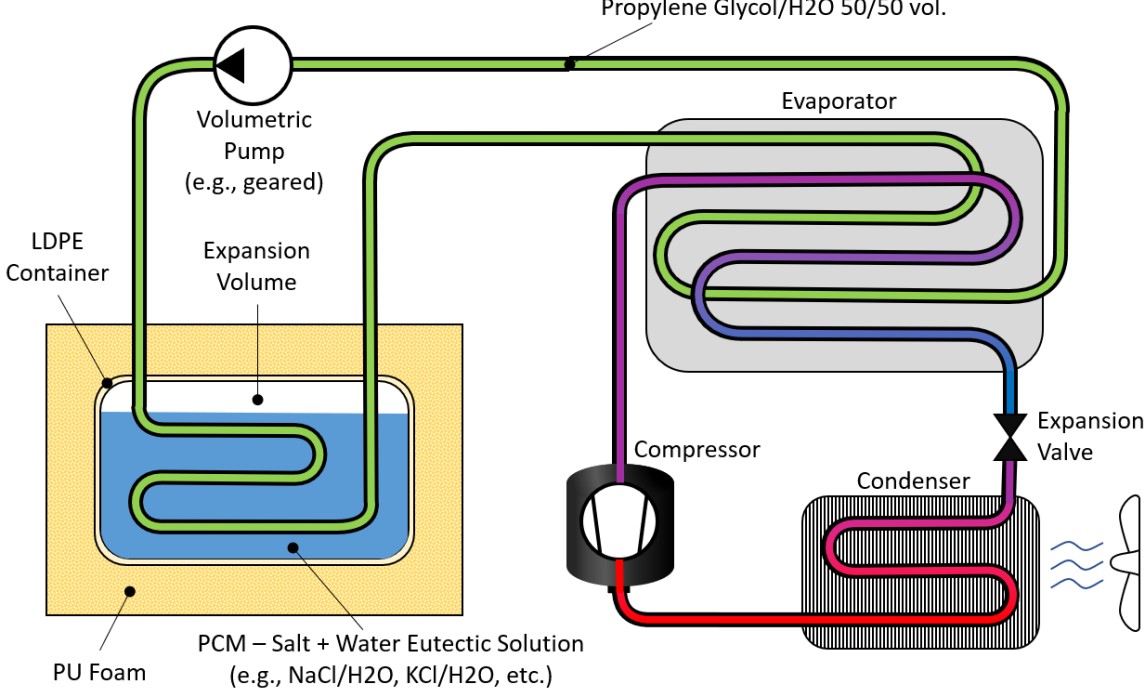

**Figure 10.** Concept of an active PCM cold storage system for a refrigeration/freezing appliance [57].

With this system, it is also possible to double or triple the amount of PCM (and consequently, the cold storage capacity) when compared with the passive approach. The disadvantages are the significantly higher cost and the detrimental effect of hydraulic (head) losses on the overall efficiency.

### 2.6. Summary: Technological Feasibility, Results and Cost of Production

Numerous approaches for the application of PCMs to domestic refrigerators have been investigated for various purposes: enhancing energy efficiency, improving temperature stabilization, increasing autonomy and resilience to power shortages. From this ample body of research, the technological feasibility was unequivocally demonstrated.

However, only a few studies investigated the benefits of PCMs for load-shifting and DSLM. Among these notable exceptions, Taneja et al., 2013, Sonnenrein et al., 2015, Barzin et al., 2015, and Maiorino et al., 2020, can be mentioned [1,8,9,23]. Despite their relative rarity, these works have also clearly demonstrated the merits of the concept, with up to 35% reduction in running costs and up to 6 h of load-shifting potential [8,23,37].

The implementation of PCMs in refrigerators inevitably implies an increase in production costs. However, this is, essentially, an industrial question, outside the realm of scientific literature. Exact figures are difficult to estimate, since they depend on a multitude of factors: type of implementation (e.g., passive vs. active), type and quantity of raw materials, degree of process automation, the scale of production, etc. Outside an industrial context, reliable quotations are also difficult to obtain. Microcontroller-based electronics are quite inexpen-

sive nowadays. The price per unit should fall in the range of USD 10.00 to USD 20.00, which considerably decreases for high production volumes. Kornbluth et al. (2012) estimated a cost of USD 50.00 per unit for a passive-type PCM application to a domestic freezer. This figure seems excessive for this type of configuration. The solution shown in Figure 9, for instance, should cost significantly less than that. The cost of raw materials—polypropylene sleeve, gellified saline solution, and adhesive—is almost negligible in the context of the total bill of materials (BOM). The PCM (NaCl solution, in this case) can be made "in-house". The main cost, in this case, should be workmanship. Additionally, this can be significantly reduced with automation. The manufacturer in question is developing a machine to apply the PCM sleeves automatically (similar to the machine that wraps the evaporator tube coil around the cold box). A solution like this should cost no more than USD 10.00 per unit. All-in, including electronics, the cost should not surpass USD 20.00 per unit. On the other hand, an active system—such as the one represented in Figure 10—will cost considerably more. Rough estimates point to a range between USD 40.00 to USD 80.00 per unit, depending on the complexity of the industrial solution. Hence, lacking a better reference, USD 50.00 can be considered a reasonable average cost for all possible configurations.

## 3. Smart Grids: Motivations, Goals and Paradigms

The essential aim of grid management is to match supply and demand. SG technologies help to achieve this goal by allowing control from the demand side (which is impossible to attain with conventional/classical control models, which rely exclusively on supply modulation). SGs incorporate the more recent advances in information and communication technologies (ICT) to gather and consolidate the relevant information from all the players in the energy chain—from producer to consumer—for accomplishing the following objectives [6,63,64]:

- Better integration and more economical, holistic, and rational management of distributed production (i.e., including distributed micro-generation resources);
- Curb GHG emissions by increasing the penetration of RES, improving their use, and minimizing production waste;
- Overall improvement of the quality of the energy supply (i.e., better voltage and frequency stability, better reliability, and consistency);
- Development and incorporation of demand response and demand-side resources, enhancing overall energy efficiency (essentially by reducing losses due to overloading of lines and transformers) and avoiding the need for fossil-fueled backup facilities;
- Enable the active participation of all consumers in the vital load balancing function (including small residential consumers);
- Implementation of real-time automated technologies that optimize the operation of appliances and consumer devices;
- Fully exploit the use of the latest information and control technologies to improve overall reliability, security, and efficiency of the energy infrastructure;
- Integration of advanced electricity storage and peak-shaving technologies, including plug-in and hybrid EVs, and residential thermal energy storage (e.g., HVAC and refrigerators/freezers);
- Better interoperability between transport and distribution networks (thanks to enhancements in communications and automation).

### 3.1. The Urgency for Smarter Grids

All the players in the energy sector—producers, transport and distribution owners, managers, governmental and non-governmental regulators, consumers, and their associations—consensually recognize the urgent need of acting on several challenging problems, affecting the present and future of the global electrical infrastructure [65]. Some of the main motivations leading toward the technological advancement of electrical grids are [6]:

1. **Intermittency and fluctuations due to wind and solar generation:** Wind and solar production are characterized by large (often unpredictable) fluctuations [66]. This caps the share of these RES, since there is a need for extra controllable and rapid response generation capacity (i.e., thermal and hydro, usually), and/or energy storage facilities (e.g., pumped hydro, electrochemical batteries, thermal phase-change, flywheels, etc.), and/or DSM or peak load shifting [67,68], to assure the required stabilization [69–71].

2. **Climate change:** The need to respect the agreed emission reductions (the Paris Agreement and global political pressures towards the reduction of $CO_2$ emissions) is pushing the investment in renewables, with their associated problems for the current infrastructures [72]. In this context, the implementation of SG paradigms becomes indispensable.

3. **Rising energy prices:** The short, medium and long-term rising trend of oil and gas prices (due to diminishing reserves, conflicts, politics, instabilities, and market fluctuations) are among the strongest motivations supporting the increasing penetration of RES and efforts to enhance energy efficiency [73,74]. The recent invasion of Ukraine by Russia has caused an unprecedented spike in oil prices, which introduced additional instability and unpredictability in oil and gas prices.

4. **Ageing, degradation, and obsolescence of infrastructures:** Capacity limitations, high losses, lack of reliability, and high maintenance costs are some of the problems associated with old infrastructures (some of them date back to the 1950s and are still in full use); the adoption of SG paradigms will optimize the usage of the existing resources, thus curbing the upgrade costs and allowing more time for updating and/or replacing the degraded infrastructures [75].

5. **Increase in global energy demand:** The existing infrastructure is being overloaded by the increase in energy demand, particularly in developing countries; this leads to a lack of reliability, quality, and consistency of the supply. Despite the significant drop in 2020, demand is still about 9% higher than in 2010 and is projected to continue rising [26,76,77]. Without the adoption of SG management models—actively involving the consumer—it will be very difficult (or even impossible) to respond adequately to this growth by investing solely in physical infrastructures.

6. **Inefficiencies resulting from the need for backup generation capacity:** The large demand fluctuations, and the lack of synchronism between the demand and renewable production, lead to the underuse of the installed generation capacity, resulting in a considerable financial burden (i.e., the backup generators are inactive most of the time, being only used in very limited peak periods) [12].

7. **Increasing distances between production, storage, and consumption sites:** RES production centers (i.e., hydro, wind, and solar) are often placed far away from the population and industries, requiring long transport infrastructures. This diverts financial resources that would be, otherwise, applied to the maintenance, renovation, and upgrade of current facilities. This also increases transport and distribution losses [78]. Distributed production and peer-to-peer (P2P) energy trading can attenuate these issues, but can also exacerbate the challenges for efficient grid control [78–80].

8. **The growing profusion of small independent energy producers:** Small production installations (i.e., mini/micro-generation) of solar photovoltaic, wind, hydro, biomass, co-generation, etc., increase the difficulties in balancing the grid from the supply side, increasing the need for innovative communication and distributed control models and technologies [79,80].

9. **New energy consumption patterns**: Energy consumption patterns have been changing considerably in the last decade, particularly in the residential sector [81]. The increasing profusion of electrical plug-in vehicles (EVs) [82], smart homes, and smart buildings offers new opportunities for control from the demand side, including load control and energy storage—e.g., charging EVs and home battery banks exclusively in off-peak hours, storing latent heat in HVAC systems, etc. [83–85].

10. **Energy market liberalization:** The liberalization of the energy markets in developed countries increased the need for energy price transparency [86,87]. This has

led to the adoption of technologies such as smart metering [88], which opens additional opportunities for implementing smarter models for DSLM by facilitating the collection of relevant consumption information and—in some more sophisticated implementations—allowing bidirectional communication.

11. **Expanding transnational grid interconnections:** To best balance supply and demand—particularly with the increasing penetration of RES—transnational grid interconnections are becoming increasingly important [89–92], introducing additional control challenges and communication issues [93–96].

### 3.2. Increasing the Penetration of RES: Challenges to Grid Control

Matching production and demand is critical for the stable operation of electrical grids. Unfortunately, wind and solar generation are infamous for their lack of consistency: they cause significant fluctuations in production, which are frequently out of phase with the demand [6,97].

This could, theoretically, be solved by implementing enough storage capacity. Regrettably, large to medium-scale storage solutions (e.g., pumped hydro, thermal storage, and lithium battery banks) are very expensive and/or complex to implement [60,98]. Consequently, they currently still have a relative residual effect [99]. This results in the necessity of keeping fossil fuel-fed power plants, for the sole purpose of stabilizing and regularizing the grid voltage [100].

In the limit situation, the installed capacity of fossil fuel-fed generators needs to be matched to an equal capacity of wind and solar production, resulting in significant economic and environmental impacts. Moreover, during peak production and low demand, most of the energy produced by these RES is not even injected into the grid, leading to significant underuse of their capacities [97].

Balancing production with demand is most efficiently performed from the demand side since—among other advantages—this approach lessens the burden on the transport and distribution infrastructure. Thus, considering the inevitable increase of RES in the energy mix, the active participation of the smaller domestic consumers in the regularization of the grids is increasingly being seriously pondered [101,102].

### 3.3. Demand-Side Load Management Paradigms

The expression "Demand-Side Management" was coined, in the early 1980s, by the Electric Power Research Institute (EPRI), being defined as (paraphrasing): "planning, implementation, and monitoring of electrical networks with the objective of influencing the consumer to use electricity in a manner that produces desirable changes in the network load profile, i.e., reducing the magnitude of its fluctuations" [103–105].

This comprises all the strategies and models aiming at influencing the consumption profile in the direction of transferring load from peak to off-peak periods [67,105]. The greatest virtue of this approach resides in the fact that it is significantly less costly to manage the load profile from the consumer side than to build a new power plant and/or increase the carrying capacity of the transport and distribution infrastructure [104–106].

DSLM is especially decisive in the context of increasing energy production from RES since it can provide a means of mitigating the supply and demand imbalances caused by the inherent fluctuating and unpredictable character of some of these energy sources. Particularly, solar and wind intermittency constitutes the main obstacle to the growth of their share in the energy market [7,107].

Since the fundamental actors in DSLM are the consumers, these need sufficient motivation and incentive to incur the inevitable costs that the installation and operation of DSLM functionalities might imply, i.e., some financial advantage needs to be offered to the energy customers by the energy providers.

### 3.3.1. Price-Based Demand Response (PBDR)

One way of achieving a more rational load profile is by implementing price models that reflect the relation between demand and supply, thus incentivizing the implementation of load management (LM) methods and technologies by consumers [108,109]. This is the most common approach adopted by energy suppliers to motivate consumers to modify their consumption patterns, in a way that helps balance the grid, by attenuating demand peaks and, thus, minimizing costly interventions from the supply side [110–112].

1.  **Time-of-use pricing (ToU):** This is the currently most practiced variable price regime in the majority of countries since it is the most straightforward to implement: it defines differentiated rates for two or three daily periods, e.g., peak, mid-peak and off-peak. It is relatively trivial to implement a cooling equipment control firmware for managing the consumption in the context of this type of pricing schedule, minimizing the energy bill [113–115].

2.  **Critical peak pricing:** According to this pricing scheme, a normal rate (usually belonging to the previously described ToU family) is valid most of the time, during the year [110]. However, during known occasions with exceptionally high demand (or, conversely, low supply), a higher rate is applied. These periods happen only for a few days, or even a few hours, during the year [110]. For the consumer, the advantage of this scheme is that the regular rate can be kept lower than it would be possible otherwise. However, that is not, ultimately, the goal: the main objective is to curb the demand by motivating the client to consume less during the short critical high-demand/low-supply periods [115].

3.  **Real-time pricing:** In this pricing system, the energy supplier broadcasts to the consumers, continually, the constantly varying energy rate. This quotation indirectly represents the continually updated relation between supply and demand, as created by the market dynamics. Energy producers dynamically increase the energy price during high-demand/low-supply periods, discouraging consumption during these critical periods [115,116]. In an SG framework, an automatism can, conceivably, be implemented to react instantly to continuously updated price fluctuations, using all available controllable resources (i.e., manageable loads and energy storage devices) to reduce the consumer energy bill and, thus, indirectly help to balance supply and demand [101,110,115–117]. Despite their designation, current implemented plans are not truly/strictly "real-time". There are, presently, two main modes of so-called "real-time" pricing being practiced: one broadcasts the quotations a day in advance (DA-RTP) and the other gives the hourly price within 60 min in advance (RT-RTP).

### 3.3.2. Incentive-Based Demand Response (IBDR)

This class of client-side energy demand modulation is based on incentives that are not directly connected to variable prices or quotations [118,119]. A contract is established between supplier and client, according to which the latter promises to respond to supplier demands for load restraints, depending on contingencies. In turn, the supplier awards the client with financial incentives (e.g., lower energy rates and/or one-off discounts) [118,119].

### 3.3.3. Direct Load Control (DLC)

In this system of demand control, the energy supplier directly controls the loads from the client side. The energy supplier benefits from an augmented ability to balance supply and demand, and the client receives financial rewards (e.g., lower rates and/or discounts) [110].

For larger clients, the supplier usually installs the necessary telemetry and telecontrol hardware at the client's site [115,120,121]. For domestic consumers, it is possible to implement direct control of appliances by factory-equipping them with electronic control units (ECUs) with appropriate firmware and Internet connectivity. Validation can be achieved with an Advanced Metering Infrastructure (AMI) (i.e., "smart meters"). Currently, pro-

grammable communicating thermostats (PCTs) have been successfully implemented in some regions.

### 3.3.4. Demand Bidding and Buyback

In these programs, a sophisticated automated negotiation protocol is established between energy suppliers and consumers. They both agree on the price—on a case-by-case basis and according to market fluctuations—for a predetermined quantity (or package) of voluntary cuts from peak-load demand made available by the consumer to the supplier [119,122,123].

These programs are currently being tried, with some success, with large consumers (e.g., industry, large commercial and office buildings). There is no fundamental obstacle to implementing these schemes for small residential consumers, although it is certainly more complicated and might prove to be, ultimately, impractical.

### 3.3.5. Emergency Demand Response (EDR)

In extraordinary cases of insufficient production and/or excess demand, energy producers set up a contingency program that includes communication with clients who can cut their load immediately, i.e., clients with energy storage facilities, backup generators, or the ability to postpone or throttle loads down—e.g., water heaters, HVAC, etc. [124].

The success of this approach depends on the ability of the supplier to predict such critical periods with enough antecedence, and on the existence of an open and effective communication channel between suppliers and consumers. This channel should be ideally telematic/digital, and the response from the demand side should be also automated, e.g., a group of electrical loads is deemed non-essential and is shut down during critical periods. Alternatively, variable power loads can be throttled down [125].

The fundamental difference between this modality and DLC is that the consumer is the one responsible for controlling the loads and not the supplier. Like with other programs, the client is rewarded with financial benefits [115].

### 3.3.6. Distributed Frequency Regulation Services

Consumers who adhere to this type of program help the suppliers maintain the stability of the AC frequency. Clients allow the supplier to directly modulate their loads, reacting quickly—in real-time—to frequency fluctuations [126]. This implies the implementation of DLC, i.e., by installing specialized equipment on the client's site, for frequency monitoring, telemetry, and load control. Again, the client receives incentives or financial compensation [110,118,119].

### 3.4. Practical Models of Load Management for the Household

In practice, the involvement of domestic consumers in DSLM can be classified in three levels, in increasing order of SG integration:

1. **ToU**—Load shifting based on fixed time-of-use pricing differentiation: the consumer adheres to a pricing scheme where the supplier defines two or three fixed daily periods with differentiated tariffs—e.g., peak, mid-peak, and off-peak (these periods can change according to the season). Based on this, the consumer schedules the consumption to minimize the energy bill. "Smart" appliances (including refrigerator-freezers) can automatically (on a time-programmed basis) decide to postpone consumption to off-peak periods, whenever possible. In this "low-level" scheme, there is no need for continuous communication between consumers and suppliers. Thus, this scheme might not be considered entirely within the realm of an SG (it might be classified as "SG level zero"). However, it still implies the use of smart meters that can record the consumption that occurs within the two or three defined periods; this approach implies no costs from the energy provider side: the consumer can simply configure the time-programming control of the appliance with the ToU periods;
2. **RTP**—Load shifting based on "real-time" forecasted pricing: instead of fixed daily periods, the supplier quotes the energy price to the consumer 24h to 1h in advance

(usually an hourly price, but it can also be in fractions of an hour). The supplier can forecast future supply and demand based on various predictable factors (e.g., weather forecasts, which determine RES production). Since it is humanly impractical for the typical domestic consumer to manually manage the load optimally in this scheme (particularly for short in-advance periods), the process should be automated via "smart appliances" with IoT connectivity, allowing these to continually receive the real-time pricing from the energy supplier and algorithmically decide when to postpone consumption; costs from the provider side are relatively negligible: it only needs to broadcast the price information via the Internet and charge the client accordingly, based on the time-referenced consumption recorded by the smart meter (assuming one is already installed);

3. **DLC**—Load management based on direct appliance control by the energy supplier: in this high-level SG scheme, the consumer grants the energy supplier direct control of some "smart appliances". The simplest implementation of DLC for the domestic consumer is the "remotely controllable HVAC thermostat" (PCT). The extension of DLC to other appliances (including refrigerator-freezers) implies the implementation of more sophisticated bidirectional IoT protocols (which implies an initial investment in information systems by the energy provider, although of relatively low financial impact). The energy supplier can use this facility to aid in load balancing, frequency regulation and emergency demand response (EDR). To make this economically advantageous for the consumer, the energy supplier can offer attractive discounts and pricing, allowing the consumer to reach a short break-even period on the required initial investment.

*3.5. Smart Grid Enabling Technologies*

A host of new technologies enables the participation of small residential consumers in SGs: Internet-of-things (IoT), smart meters (SMs) and advanced metering infrastructures (AMI) [88,127,128], home automation (HA), home area networks (HAN), neighborhood area networks (NAN), wide area networks (WAN) [129], meter data management systems (MDMS), home energy management systems (HEMS) [101,112], and building energy management systems (BEMS), just to mention the more relevant ones [110,130,131].

These technologies provide a workable environment for DSLM/DR programs, harmonizing supply and demand via the modulation of consumption. The participating consumers receive financial incentives from the energy suppliers, rewarding their participation in these programs [110,118,119].

In a Communication-Based Demand Response (CBDR) program, household appliances communicate with SMs, establishing an automated bidirectional communication between domestic consumers and energy suppliers. Consumer appliances receive updated information about energy prices—in real-time or not—and can, autonomously, make load management decisions, using their capacity to delay consumption and/or store energy (thus minimizing the electricity bill). Reciprocally, appliances can offer their available capacity for on-demand immediate consumption to the energy marketplace, where they can be commanded by suppliers [132].

3.5.1. Advanced Metering Infrastructures (AMI)

One of the assets that eases the evolution toward smarter electric grids, is the growing installed network of SMs in developed countries. These apparatuses contain a microcontroller, memory, updatable firmware, and means of remote communications with the energy suppliers (generally via PLC and/or GSM) [127,128,133].

The American Federal Energy Regulatory Commission (FERC) defines Advanced Metering Infrastructure as "meters that measure and record consumption data, hourly or with greater frequency, supplying this information to both users/clients and energy suppliers, at least daily, that information being used for billing and other purposes" [110].

SMs are essential for the implementation of SGs [88,134]. Their most basic function is to record daily consumption history, allowing ToU billing. Nevertheless, they can also offer several other functions in the context of DSLM/CBDR [102,135]. For example, these apparatuses can serve as a communications gateway with electrical loads that can be remotely controlled. They can also supply vital real-time information about where consumption is taking place. Importantly, they can also validate both energy saving and/or load shifting actions by the consumers [136].

### 3.5.2. Wide Area Network Infrastructures

A fundamental requirement for the implementation of SG functionalities is the availability of communication means between all the players (i.e., production, transport, control, distribution, and consumption). Fortunately, nowadays, wide-area wireless Internet access technologies are globally widespread. Güngör et al. (2011), Gao et al. (2012), Usman and Shami (2013), Supriya et al. (2015), Mahmood et al. (2015), and Emmanuel and Rayudu (2016), conducted very comprehensive reviews of available communication technologies and standards for SGs [130,131,137–139], which are still broadly applicable and up to date.

1. **Mobile cellular infrastructure:** The widespread coverage of cellular infrastructure is a valuable resource for supplying the necessary means of communication for the implementation of SG paradigms. Given the relatively modest bandwidth needs of these applications, even the most basic GSM/2G coverage can be sufficient for this purpose [130,131,137]. Many SMs are equipped with GSM connectivity.
2. **Power Line Communication (PLC):** Since all the constituent parts of the electrical grid are, inherently, physically interconnected by conductor wires, it is logical to use this pre-existing infrastructure as a communication medium. This would have the advantages of not depending on third-party infrastructures and allowing an economic means of communication between all grid constituents. Frequently, SMs are equipped with PLC transceivers. Plentiful technologies, protocols, and standards exist for this purpose—e.g., IEC 14908-3 (Lon Works), IEC 14543-3-5 (KNX, BUS), CEA-600.31 (CEBus), IEC 61334-3-1 (DLMS), IEC 61334-5-1, IEEE 1901.2, ITU-T G.henm, PRIME, G3-PLC, IEEE 1901, TIA-1113 (HomePlug 1.0), ITU-T G.hn (G.9960/G.9961), and HD-PLC [130,131]—but their implementation is still relatively limited and of local/regional scale.

### 3.5.3. Internet-of-Things

According to the Recommendation, ITU-T Y.2060 of the Telecommunication Standardization Sector of the International Telecommunications Union (ITU), the IoT "( … ) can be viewed as a global infrastructure for the information society, enabling advanced services by interconnecting (physical and virtual) things based on existing and evolving interoperable information and communication technologies (ICT)" [140].

It is unviable to list all the protocols encompassed under this concept: they are numerous and still growing (e.g., ZigBee [141]). The relevance of this definition in the current context is justified by the fact that the exchange of information between SG devices will adopt some of these protocols [36,107,142]. Furthermore, the application of IoT technologies for real-time demand response in the household, specifically narrowband-IoT (NB-IoT), is presently being investigated [143], particularly in the context of the newest SG paradigms [144,145].

### 3.6. SG and IoT Standard Protocols

The interoperability between the several types of equipment involved in the energy transmission chain—from production control and distribution infrastructure management to energy metering and consumer apparatuses—presupposes a "common communication language", i.e., compliance with standard telematic protocols [130]. In the specific realm of SGs, several standards have already been independently defined in various parts of the world (e.g., the US and Europe). Some of these were, fortunately, unified under

the International Organization for Standardization (ISO). Along with these SG-specific communication standards, general-purpose IoT protocols are also relevant in the current context, particularly for communication with domestic appliances.

### 3.6.1. SG-Specific Protocols

#### Open Smart Grid Protocol (OSGP) (ISO/IEC 14908)

The European Standards Organization (ESO) was mandated by the European Commission (EC) with the task of defining standards and norms for smart electricity meters in March 2009, and for EVs in June 2010 [72]. With the consensus of all participants in the Smart Grids Task Force, the EC emitted a directive for the standardization of SGs [72].

The European Telecommunications Standards Institute (ETSI) was created in 1998 by the European Conference of Postal and Telecommunications Administrations (CEPT) for the definition of telecommunication industry standards in Europe. It was officially recognized by the EC and by the European Free Trade Organization secretariat. In January 2012, ETSI published the first version (V1.1.1) of the Open Smart Grid Protocol (OSGP) [146]. In December 2016, the second version (V2.1.1) was published and, finally, in January 2019, the first (and last, so far) revision (V2.2.1) was published.

OSGP is very comprehensive and detailed: the most recent specification document comprises 252 pages. With more than 30 million OSGP-compliant meters equipped with PLC installed in Europe [146], OSGP may eventually become a global de facto standard. Used in conjunction with the international networking standard ISO/IEC 14908 (which includes IP over powerline) [147], it currently provides the most wide-ranging technological framework for the implementation of SG solutions [145].

#### LonTalk/LonWorks (ISO/IEC 14908)

LonTalk is a networking protocol originally developed by the American company Echelon Corporation which is part of the technology platform called LonWorks [148]. This standard specifies a multi-purpose control network protocol stack, optimized for SGs, smart buildings, and smart city applications. It operates over transmission media such as twisted pairs, powerlines, fiber optics, and radio (RF). It is currently in widespread use for industrial control, home automation, transportation, and building systems (such as lighting and HVAC). The protocol was included in the open international control networking family of standards ISO/IEC 14908 [147], under the ISO/IEC JTC 1/SC 6 norm [149].

In 2007, the European Committee of Domestic Equipment Manufacturers (CECED), adopted LonWorks as part of the Household Appliances Control and Monitoring Application Interworking Specification (AIS) standards [36].

#### Open Automated Demand Response Communications Specification (OpenADR)

OpenADR is an American SGs protocol sponsored by the US Department of Energy Office of Scientific and Technical Information (www.osti.gov) and subsequently submitted to the Institute of Electrical and Electronics Engineers (IEEE). The development of OpenADR began in 2002 (independently from LonTalk/LonWorks) by the Demand Response Research Centre (DRRC), supervised by the Lawrence Berkeley National Laboratory (LBNL) [150]. The specification defines "an open standards-based communications data model designed to facilitate sending and receiving demand response, price, and reliability signals from a utility or Independent System Operator to electric customers" [150]. Its intent is "to provide interoperable signals to building and industrial control systems that are preprogrammed to take action based on a demand response signal" [150]. Thus, the objective is to enable the automation of DR, based on real-time information—e.g., on the energy market—or specific commands or requests issued by the utility operators to the clients [151].

### 3.6.2. IoT Protocols

ECHONET Lite

Originating in Japan, ECHONET Lite is an IoT protocol, developed by a wide consortium of Japanese private companies, universities and institutes (about 200 in total) [152]. Its objective is the establishment of communication with and between household appliances [153]. According to the consortium's stated goals, ECHONET is a communication protocol designed to create the "smart houses" of the future (http://echonet.jp/english, accessed on 30 August 2022). The protocol specification encompasses the communication with SMs, allowing for an intelligent electrical load modulation in a DSLM context [153,154].

Despite the consortium's ambitions, this protocol is currently confined to Japanese branded home appliances and, so far, exclusively destined for the Japanese domestic market.

ZigBee (IEEE 802.15.4)

Due to its increasing popularity and profusion of compliant appliances—currently about 3500 certified and about 300 million deployed worldwide, according to the ZigBee Alliance (and even on Mars!)—there is an abundant body of research focusing on the potential use of this relatively recent wireless IoT protocol suite for the implementation of domestic SG functionalities, specifically advanced metering (AM) and DSLM [130,137,155–162].

This open high-level communications protocol suite was developed in the early 2000s by the ZigBee Alliance (zigbeealliance.org) as a "very low-cost, very low-power-consumption, two-way, wireless communication standard" [141]. It is based on the IEEE 802.15.4 standard for "low-rate wireless networks" [163]. The protocol is especially adequate for the establishment of wireless personal area networks (WPANs) and/or HANs. With its multi-node, wireless mesh capabilities, it is much more versatile than other wireless protocols (e.g., Bluetooth®), while its specification is more complete and application-oriented than other more general ones (e.g., Wi-Fi). Being bi-directional makes it particularly adequate for HA and, consequently, for LM and DR in the domestic environment. An important feature in this context is its inherent security, thanks to the implementation of 128-bit symmetric encryption keys [141].

### 3.7. Other Relevant Protocols and Standards

It is not possible to list every standard and norm in the current context. Nevertheless, the list would not be complete without mentioning the pervasive Bluetooth and Wi-Fi (IEEE 802.11) wireless protocols.

Other standards worth mentioning include 6LoWPAN, Z-Wave, WiMAX (IEEE-802.16), DASH7 (ISO/IEC 18000-7), and Device Language Message Specification (DLMS) [130,131,137], IEC 14908-3 control network protocol, power line channel specification; IEC 14543-3-5 home electronic system (HES) architecture, media, and media-dependent layers, powerline for network-based control of HES Class 1; IEC 61334-3-1 distribution automation using distribution line carrier systems, mains signaling requirements, frequency bands, and output levels; IEC 61334-5-4 distribution automation using distribution line carrier systems, lower layer profiles, multi-carrier modulation (MCM) profile; CEA 600.31 power line physical layer and medium specification; and KNX, standardized (EN 50090, ISO/IEC 14543) OSI-based network communications protocol for intelligent buildings [130,131]. Table 4 summarizes some of the most relevant SG and IoT protocols in current use for DSLM.

### 3.8. Summary: Current State-of-the-Art and Prospects for the Future

Given the present pressures (development, climate change and energy crisis), the adoption of technologies for more rational management of the electrical grid is urgently needed. This implies the active participation of all energy consumers, regardless of their dimension. At the level of domestic consumers, the ubiquitous refrigerator is one of the most promising resources for DSLM. Even without the use of PCMs, several studies claim potential results of up to 37.9% in load shifting, 5% in global peak-load shaving, and a 51% reduction in running costs [1,12,18].

**Table 4.** SG and IoT communication protocols.

| Class | Protocol | Standard(s) | Organization(s) | Reference(s) |
|---|---|---|---|---|
| SG | Open Smart Grid Protocol (OSGP) | ISO/IEC 14908 | ESO, ETSI, CEPT, ISO, IEC | [72,145,146] |
| | LonTalk/ LonWorks | ISO/IEC 14908 JTC 1/SC 6 | Echelon Corp., CECED, ISO, IEC | [36,148,149] |
| | OpenADR 2.0 | | LBNL, DRRC, USDE, OSTI, IEEE | [150,151] |
| IoT | ECHONET Lite | | A consortium of Japanese Companies, Universities, and Institutes | [152,153] |
| | ZigBee | IEEE 802.15.4 | ZigBee Alliance, IEEE | [130,137,141,157–163] |

Gauging from the above survey, a sufficient and comprehensive framework of technologies and ICT protocols has already been established for the implementation of all modalities of DSLM for the domestic consumer—from the most basic ToU tariff schemes to the most advanced forms of DLC. Most of the investments in the essential advanced metering infrastructure and communication networks have also already been made in the "developed world". The inclusion of the domestic energy market in SG paradigms is, therefore, not currently hampered by a lack of technological solutions but, possibly, by some inertia and lack of vision from policy and decision-makers. A determined intervention from the public sector is essential to motivate the energy industry to undertake the needed steps and investments. Regulatory frameworks, new energy-efficiency labelling for electrical appliances, financial incentives to energy providers and consumers and public awareness campaigns are among the possible public intervention avenues.

Regarding implementation costs from the supplier side, they should be—theoretically—relatively insignificant, if an advanced metering infrastructure has already been put into place. In the most basic ToU scheme, the consumer simply programs the ToU periods into the appliance (i.e., zero cost for the supplier). In an RTP regime, the supplier simply needs to broadcast the pricing information via the Internet (with negligible costs). With a DLC model, the supplier needs to implement a relatively more sophisticated information system. Costs associated with the latter will be higher than with the two former schemes but, still, relatively insignificant in the context of the budget involved in building and maintaining production, transport, distribution and metering infrastructures. In all cases, the energy provider charges the consumer according to the time-referenced consumption history recorded by the SM.

## 4. Discussion

### 4.1. The Inevitability of the Adoption of SG Paradigms

Considering the numerous and significant motivations, the adoption of SG technologies is virtually inevitable. Supported by numerous practical, logistical, and economic arguments, this thesis is consensual in the electrical engineering community. It is also reinforced by the growing profusion of publications on the subject [164]. Considering this convergence of viewpoints and interests, and the current availability of enabling technologies—particularly in the last two decades, with the advancement and increased establishment of digital communication standards and infrastructures.

### 4.2. Hurdles Facing the Implementation of SGs

It is legitimate to raise the question of why there has not been a more significant advancement in terms of the practical implementation of SG concepts. The answer might partially lie in the practical difficulties of establishing consensuses about the ideal modes of implementing dynamic and transparent energy marketplaces, involving the active

participation of all final consumers including small residential ones, and, more specifically, about the establishment of legal and contractual support frameworks [165,166].

Another significant obstacle is the absence of a truly universal agreement about the indispensable communication protocol standards. Without a broad consensus, the advancement of distributed SG frameworks, involving all relevant players, will be significantly hampered. On the positive side, although undertaken separately, the European and American initiatives (OSGP and LonTalk/LonWorks, respectively), were united under the common umbrella of an ISO standard (ISO/IEC 14908, 2012), thus providing an important degree of standardization and interoperability. On the negative side, other national-scoped initiatives (e.g., *ECHONET*) have failed to integrate and, due to some fundamental differences, the future does not look promising for such an integration.

### 4.3. Reasons for Optimism

The growing body of literature focusing on the topics of TES, LHCS, and PCMs denotes the potential importance of the role that these approaches may play in a smarter global energy management framework [50,53,167,168]. From the published research and statistics, the potential impact of using household refrigerator-freezers for DSLM/DR is sufficiently significant and not to be ignored. For this application, the use of PCMs holds great promise, as they can, potentially, enable the storage of latent cold when energy is abundant and/or cheap, thus providing enough storage to avoid consumption during peak demand and/or low-supply periods [45]. Ideally combined with SGs connectivity, this ability will allow more efficient exploitation of the potential of intermittent RES (e.g., wind and solar).

### 4.4. State-of-the-Art of Enabling Technologies

All enabling SG technologies have reached maturity and relatively widespread adoption, particularly in what concerns information and communications technologies (ICTs).

Smart metering (SM) infrastructures have reached a sufficient level of completion in some regions—particularly in Europe and North America—to enable the most advanced models of DSLM, including direct control of household appliances by the energy suppliers in real-time, augmenting their grid stabilization capabilities.

### 4.5. Feasibility of Using Domestic Refrigeration for Load Shifting

Technologically, the industrial implementation of LHCS with PCMs in home refrigerators/freezers is, demonstrably, feasible. Corroborating this conclusion, some refrigerators equipped with PCMs were already tried on the market and others are being currently developed [61,62]. Numerous PCM types and application configurations have already been investigated and/or implemented, either passive or actively controlled [169]. From the point of view of production costs, the passive application of inexpensive eutectic saline solutions in thermal contact with the evaporator is the most cost-effective. The added thermal inertia can be readily harnessed for load shifting. Even in a disconnected context (i.e., outside an SG framework), deferring the load to off-peak periods can result in appreciable energy bill savings for the consumer [8,9,13,23].

### 4.6. Economic Viability

Even without implementing LCS-enhancing technologies like PCMs, the inherent thermal inertia of household refrigeration and freezing appliances enables them to be used for DSLM, with minimal added manufacturing costs. Most of these apparatuses are already microprocessor controlled. Load-shifting capabilities could be added by simple firmware implementation. Appliances including IoT connectivity (like Wi-Fi and ZigBee) are also becoming more common. This connectivity allows the implementation of advanced SG approaches like RTP and even DLC, with virtually no added production costs (again, by simply programming them into the firmware).

As previously stated, the application of PCM technologies can enhance the LCS capacity of these appliances, greatly augmenting their load-shifting capabilities. Some

estimations calculated that the impact on manufacturing costs will be around USD 50.00 per unit (or even lower, according to the authors' estimations) [13]. The increase in initial investment can be, however, quickly amortized by the consumer through savings in the energy bill. Considering current energy costs, break-even can be achieved within less than a year—in the best circumstances—or two years, at most [9,16,18,23].

Additionally, the systemic economic and environmental benefits (including the potential GHG emissions reduction) justify the implementation of state/public incentives, motivating the consumer option for products with these functionalities [18,20]. These can include easily implementable product labelling norms, subsidization, and/or tax abatements (e.g., income tax deductions and/or sales tax/VAT reductions).

### 4.7. Further R&D Is Still Needed

An actively controlled cold storage system will be necessary to harness the maximum potential for load shifting and demand response while also maximizing efficiencies and effectiveness [1,57]. This will impact production costs negatively, increasing the difficulty of achieving commercially viable solutions. To motivate and convince consumers, energy efficiency labelling should, ideally, reflect the cost savings that these active energy storage and load shifting solutions can provide [170,171]. Unfortunately, R&D on actively controlled cold storage with household refrigeration appliances is still very scarce. More is needed.

### 4.8. Redirecting R&D Focus to Load Shifting

Thus far, most research into the application of PCMs to domestic cooling appliances has predominantly focused on improvements in energy efficiency (e.g., by reducing the frequency of compressor on-off cycles) [50,53,167,172–183]. The potential of using the cold storage ability of PCMs for load shifting in domestic refrigerators-freezers has seldom been thoroughly investigated, particularly in the context of SGs. Notable exceptions are Taneja et al. (2013) and Waschull et al. (2014) [1,17]. Hence, the potential for producing novel science on this specific topic remains high.

### 5. Conclusions

The global imperative of reducing GHG emissions in a scenario of growing global energy demand is not possible without increasing the penetration of RES. The adoption of SG technologies is, therefore, inevitable. Due to their inconsistent nature, increasing wind and solar production will not be feasible without expanding storage capacities and augmenting electrical load management capabilities. SG technologies allow consumers to participate in the vital task of balancing supply and demand more actively and effectively. To fully maximize storage and load shifting capacities, small household consumers need to be fully engaged in this paradigm. Digital communication technologies and standards are sufficiently mature and widespread to enable this. The degree of implementation of advanced metering infrastructures has also attained the required level in several regions of the developed world. Nevertheless, further public sector intervention is important to effectively motivate consumers and energy providers to incur the required investment.

Home refrigeration and freezing appliances are particularly suitable for load shifting, as they are permanently connected to the grid. This makes them available for the implementation of advanced SG strategies, like direct load control by the energy supplier. Even without cold storage enhancing technologies—e.g., PCMs—the inherent thermal inertia of these apparatuses allows them to be used for some degree of DSLM. Globally, these appliances consume about 5% to 8% of the total electrical energy production. If fully exploited, the impact of this potential cannot be neglected in a necessarily holistic solution to climate change. These appliances are responsible for about 14% of the energy consumed by the residential sector, causing yearly emissions of about 450 million $CO_2$ tons [40,41]. If 50% of worldwide refrigerator-freezers could shift 40% of their consumption to off-peak hours or peak RES production, annual emissions of more than 100 million tons of GHGs could be, potentially, avoided.

The use of PCMs can significantly enhance the energy storage capacity of these appliances, further augmenting their load-shifting capabilities. Current research has demonstrated that this is technologically feasible and can be economically viable. Some refrigerators equipped with PCMs were already tried on the market and others are being currently developed [57,61,62]. The estimated impact on manufacturing costs will be around USD 50.00 per unit [13]. If reflected on the price tag, this cost can be quickly amortized in about a year or two, considering the potential electricity bill savings of load shifting from peak to off-peak periods, at current energy costs [9,16,18,23]. The increase in global energy demand and cost are contributing factors toward the viability of these ideas. Numerous experimental studies demonstrate that by using load shifting strategies and technologies, the residential consumer can obtain energy bill savings of approximately 15% to 60%, depending on local energy pricing and other particular conditions [9,16,18,23]. However, more research is still needed to further exploit the synergistic potential of PCMs and SG technologies in home refrigeration.

**Author Contributions:** Conceptualization, L.S.R., J.A.F. and V.A.F.C.; methodology, L.S.R., J.A.F. and V.A.F.C.; validation, D.L.M.; investigation, L.S.R. and D.L.M.; resources, L.S.R. and D.L.M.; data curation, L.S.R. and D.L.M.; writing—original draft preparation, L.S.R.; writing—review and editing, L.S.R., D.L.M., J.A.F., V.A.F.C. and N.D.M.; visualization, L.S.R.; supervision, J.A.F. and V.A.F.C.; project administration, F.J.N.D.S.; funding acquisition, F.J.N.D.S. All authors have read and agreed to the published version of the manuscript.

**Funding:** This work was supported by the R&D project UFA+EE—Research and Development of Autonomous and Energy Efficient Cold Units, financed by the European Union through its Structural and Investment Funds, and through the programs Portugal 2020, "Programa Interface", System of Incentives for Research and Technological Development (SI I&DT) [grant number 03/SI/2017]; by the FCT, the Portuguese national foundation for science and technology [grant number 2020.06120.BD] and [grant number 2021.06083.BD]; and by the projects [UIDB/00481/2020], [UIDP/00481/2020] (FCT), and [CENTRO-01-0145-FEDER-022083], under the program "Centro Portugal Regional" (Centro2020), and the PORTUGAL 2020 partnership agreement, through the European Regional Development Fund.

**Data Availability Statement:** Not applicable.

**Acknowledgments:** The authors would like to express their gratitude to Vítor Silva, and Tensai Indústria, SA (www.tensai.pt), for graciously providing access to their research data on the industrial application of PCMs to commercial cooling equipment.

**Conflicts of Interest:** The authors declare no conflict of interest. The funders had no role in the design of the study; in the collection, analyses, or interpretation of data; in the writing of the manuscript; or in the decision to publish the results.

## Abbreviations

| | |
|---|---|
| AMI | Advanced metering infrastructure |
| CBDR | Communications-based demand response |
| DLC | Direct load control |
| DR | Demand response |
| DSM | Demand-side management |
| DSLM | Demand-side load management |
| EDR | Emergency demand response |
| EIA | Energy Information Administration (of the USA) |
| EPRI | Electric Power Research Institute |
| ESO | European Standards Organization |
| ETSI | European Telecommunications Standard Institute |
| FERC | Federal Energy Regulatory Commission |
| GHG | Greenhouse gases |
| GSM | Global system for mobile communications |
| HVAC | Heating, ventilation, and air conditioning |
| IBDR | Incentive-based demand response |
| ICT | Information and communication technology |

| IEA | International Energy Agency |
| --- | --- |
| IEC | International Electrotechnical Commission |
| IEEE | Institute of Electrical and Electronics Engineers |
| IEEE-PES | IEEE Power & Energy Society |
| IoT | Internet-of-things |
| ISO | International Standards Organization |
| LAN | Local area network |
| LHCS | Latent heat-cold storage |
| LMS | Load management system |
| LM | Load management |
| OSGP | Open smart grid protocol |
| OSI | Open systems interconnection |
| PBDR | Price-based demand response |
| PCM | Phase-change material |
| PCT | Programmable communicating thermostat |
| PLC | Power line communication |
| PLS | Peak load shifting |
| RES | Renewable energy sources |
| RTP | Real-time pricing |
| SG | Smart grid |
| SM | Smart meter |
| SoC | State-of-charge (usu. in a battery) |
| TES | Thermal energy storage |
| ToU | Time-of-use pricing |

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
