# Peer review of "The Load Shifting Potential of Domestic Refrigerators in Smart Grids: A Comprehensive Review"

_energies, doi:10.3390/en15207666_

Round 1

Reviewer 1 Report

1)            Abstract

The authors of the manuscript submitted for review deal with the scientifically important problem of managing the energy consumption of domestic refrigeration appliances. The aim is to control appliances so that they use electricity when renewable sources are available. The aim and scope of the work are sound and worth investigation. This paper is review type but also contains interesting analysis. The abstract is written well.

2)            Introduction

The introduction defines the purpose of the work well and presents the principle of the system. There is also a well-shown comparative analysis of previous work in the field. I have no reservations about this presentation.

3)            Body part of paper

Methods of storing thermal energy as well as electric energy are discussed with the respect of this application. There is also a schematic drawing of the PCM application for a refrigerator, but without dimensioning which makes difficult to evaluate the solution possibility. The dimensioning and cost for individual refrigerator is the crucial element for user decision: purchase or not this solution. Therefore this element of the manuscript has to be improved.

The intelligent grid for appliance control is also presented. 

4)            Conclusion

The conclusions presented in the paper are actually mostly correct, except that a 50USD increase in the cost of a single fridge is not a total increase in operating costs. Added to this are the costs of installing and operating a smart grid. On top of that, these components introduce an additional energy cost in their manufacture and installation that is difficult to estimate. However, it can be assessed by the financial outlay and its balance. In spite of my doubts about the final estimates, I believe that the work is worth publishing as it may give rise to a broader discussion or perhaps a limited implementation.

5)            Reference

The literature review contains 183 publications on this subject which is complete in my opinion.

Reviewer 2 Report

This manuscript reviews domestic refrigeration and freezing appliances for electrical load shift (combined with phase-change materials). The workload of the manuscript is very large, but there are some problems in the basic framework and contents.

1. Framework level. There are 17 sections in the manuscript, which is too long for a paper. In the introduction, the major contributions and research gaps of the manuscript are not explained, and the contents of subsequent sections are not summarized, which will make it difficult for readers to grasp the key points of the paper.

2. Content level. The manuscript covers a wide range of contents and has certain popular science value. But for the research review, this is unqualified. A qualified review should make an in-depth summary and analysis of the research content in the field (for example, make an in-depth summary and analysis of various materials and control technologies that can be applied to the cold storage of refrigerators, explain the scientific problems to be solved, and indicate the research directions that should be carried out in the future). However, the content of this manuscript is very scattered, it gives the reader a sense of patchwork (for example, Section 2 and section 3 can be combined into the introduction. Because section 2 is a summary of existing research, and section 3 is the potential impact of household refrigerators on peak load shift, the purpose of these two sections is to discuss the necessity of research).

Reviewer 3 Report

I suggest you review the formatting of the text e.g. line 398, 403 and a few others.

It is likely that the use of reference [58] is not appropriate for a high-level journal, so I would recommend using a different reference.

Round 2

Reviewer 2 Report

Accept the revised version and agree to publish it in Energies.